# DiscoBAX: Discovery of optimal intervention sets in genomic experiment design

## Abstract

The discovery of therapeutics to treat genetically-driven pathologies relies on identifying genes involved in the underlying disease mechanism. With billions of potential hypotheses to test, an exhaustive exploration of the entire space of potential interventions is impossible in practice. Sample-efficient methods based on active learning or Bayesian optimization bear the promise of identifying targets of interest using as few experiments as possible. However, genomic perturbation experiments typically rely on proxy outcomes measured in biological model systems that may not completely correlate with the results of interventions in humans. In practical experiment design, one aims to find a set of interventions that maximally move a target phenotype via a diverse mechanism set to reduce the risk of failure in future stages of trials. To that end, we introduce DiscoBAX — a sample-efficient algorithm for genomic intervention discovery that maximizes the desired movement of a phenotype while covering a diverse set of underlying mechanisms. We provide theoretical guarantees on the optimality of the approach under standard assumptions, conduct extensive experiments in synthetic and real-world settings relevant to genomic discovery, and demonstrate that DiscoBax outperforms state-of-the-art active learning and Bayesian optimization methods in this task. Better methods for selecting effective and diverse perturbations in biological systems could enable researchers to discover novel therapeutics for many genetically-driven diseases.

## 1 Introduction

Genomic experiments probing the function of genes under realistic cellular conditions are the cornerstone of modern early-stage drug target discovery and validation; moreover, they are used to identify effective modulators of one or more disease-relevant cellular processes. These experiments, for example using Clustered Regularly Interspaced Short Palindromic Repeats (CRISPR) (Jehuda et al., 2018) perturbations, are both time and resource-intensive (Dickson & Gagnon, 2004; 2009; DiMasi et al., 2016; Berdigaliyev & Aljofan, 2020). Therefore, an exhaustive search of the billions of potential experimental protocols covering all possible experimental conditions, cell states, cell types, and perturbations (Trapnell, 2015; Hasin et al., 2017; Worzfeld et al., 2017; Chappell et al., 2018; MacLean et al., 2018; Chappell et al., 2018) is infeasible even for the world's largest biomedical research institutes. Furthermore, to mitigate the chances of failure in subsequent stages of the drug design pipeline, it is desirable for the subset of precursors selected in the target identification stage to operate on diverse underlying biological mechanisms (Nica et al., 2022). That way, if a promising candidate based on in-vitro experiments triggers unexpected issues when tested in-vivo (e.g., undesirable side effects), other lead precursors relying on different pathways might be suitable replacements that are not subject to the same issues. Mathematically, finding a diverse set of precursors corresponds to identifying and sampling from the different modes of the black-box objective function mapping intervention representations to the corresponding effects on the disease phenotype (§ 2). Existing machine learning methods for iterative experimental design (e.g., active learning, Bayesian optimization) have the potential to aid in efficiently exploring this vast biological intervention space. However, to our knowledge, there is no method geared toward identifying the modes of the underlying black-box objective function to identify candidate interventions that are both effective and diverse (§ 6).

To this end, we introduce DiscoBAX - a sample-efficient Bayesian Algorithm eXecution (BAX) method for discovering genomic intervention sets with both high expected change in the target phe-

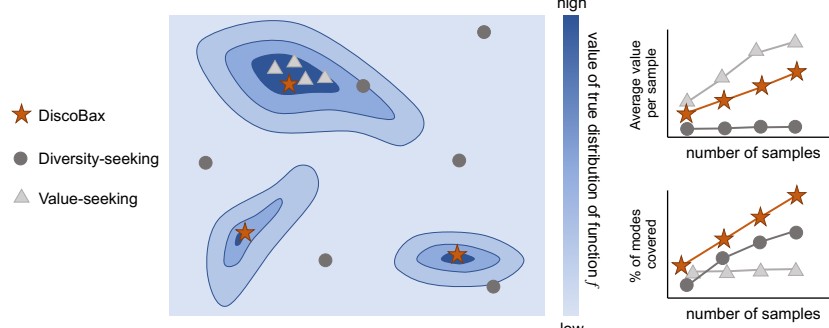

Figure 1: We compare DiscoBAX (orange star) to existing diversity-seeking (dark grey circle) and value-seeking (light grey triangle) batch active learning policies. DiscoBAX aims to recover a maximally diverse set of interventions with values above a pre-defined threshold from a given underlying distribution. This aim contrasts with value-seeking strategies focusing on maximizing value and diversity-seeking strategies focusing on maximizing coverage. We expect DiscoBAX to design genomic experiments yielding high value findings that maximize mode coverage. As discussed in § 1, the diversity of selected interventions is highly desirable to increase the chances that at least some of these interventions will succeed in subsequent stages of the drug discovery pipeline.

notype and high diversity to maximize chances of success in the following stages of drug development (Figure 1), which we formalize as set-valued maximization problem (Equation 4). After providing theoretical guarantees on the optimality of the presented approach under standard conditions, we perform a comprehensive experimental evaluation in both synthetic and real-world datasets. The experiments show that DiscoBAX outperforms existing state-of-the-art active learning and Bayesian optimization methods in designing genomic experiments that maximize the yield of findings that could lead to the discovery of new potentially treatable disease mechanisms.

Our contributions are as follows:

- We formalize the gene target identification problem (§ 3) and discuss limitations of existing methods in addressing this problem (§ 6).
- We develop DiscoBAX - a sample-efficient BAX method for maximizing the rate of significant discoveries per experiment while simultaneously probing for a wide range of diverse mechanisms during a genomic experiment campaign (§ 4).
- We provide theoretical guarantees that substantiate the optimality of DiscoBAX under standard assumptions (§ 4 and Appendix A).
- We conduct a comprehensive experimental evaluation covering both synthetic as well as real-world experimental design tasks that demonstrate that DiscoBAX outperforms existing state-of-the-art methods for experimental design in this setting (§ 5).

## 2 BACKGROUND AND NOTATION

Genomic experimentation is an early stage in drug discovery where geneticists assess the effect of genomic interventions on moving a set of disease-relevant phenotypes to determine suitable drug targets. In an abstract language, we assume a black-box function, $f : \mathcal{G} \to \mathbb{R}$, that maps each gene, $g \in \mathcal{G}$, to the value, $f(g)$, corresponding to the magnitude of phenotypic change under gene knock out. The set, $\mathcal{G}$, is finite, $|\mathcal{G}| = m < \infty$, because there are a limited number of protein-encoding genes in the human genome ($\approx 20,000$) (Pertea et al., 2018), and is formalizable by either the set of integers or one-hot vectors with dimension $m$. However, biologically informed embeddings, $\mathbf{X} : \mathcal{G} \to \mathcal{X}$, are often preferred to represent genes for their potential to capture genetic, functional relationships. We assume that gene embeddings, $\mathbf{X}(g) = \mathbf{x} \in \mathcal{X} \subseteq \mathbb{R}^d$, are $d$-dimensional variables, with $m$ distinct members, $|\mathcal{X}| = m$, thus, we use $f(g)$ and $f(\mathbf{x})$ interchangeably.

In drug development, a candidate target must meet several criteria to proceed to subsequent stages in the development pipeline. For example, engaging the target – down- or up-regulating the gene – must move the phenotype *significantly* in the desired direction. Such genes are called "top-movers" of the phenotype. We can define the $K$ top-movers for a given phenotype as members of the set, $\mathcal{X} = \{\mathbf{x}_1, \mathbf{x}_2, \ldots, \mathbf{x}_m\}$, corresponding to the $K$ largest values of $\{f(\mathbf{x}_1), f(\mathbf{x}_2), \ldots, f(\mathbf{x}_m)\}$. However, each evaluation of the phenotype change, $f$, requires a CRISPR-Cas9 knockout experiment in the lab, which makes exhaustive experimentation infeasible even for the most resourceful institutions. Hence in practice, the experimentation budget is limited to $T \ll m$ experiments. Instead of choosing the $K$ top-movers (requiring phenotype change knowledge, $f(\mathbf{x})$, for all inputs $\mathbf{x} \in \mathcal{X}$), a more practical approach is to form the subset, $\mathcal{X}_c \subseteq \mathcal{X}$, of genes that when knocked out lead to a change in the phenotype, $f(\mathbf{x})$, larger than a selected threshold value, $c$, i.e. $\mathcal{X}_c := \{\mathbf{x} \in \mathcal{X} : f(\mathbf{x}) \geq c\}$.

Bayesian Algorithm Execution (BAX), proposed by Neiswanger et al. (2021), is a method to estimate the output, $O_\mathcal{A} := O_\mathcal{A}(f)$, of an algorithm, $\mathcal{A}$, run on a function, $f$, by evaluating the function on a budgeted set of inputs, $\{\mathbf{x}_i\}_{i=1}^T \in \mathcal{X}$. Estimating a computable property is done by positing a probabilistic model for $f$ for estimating $O_\mathcal{A}$. Data is acquired by searching for the value $\mathbf{x} \in \mathcal{X}$ that maximizes the mutual information, $I(Y_\mathbf{x}; O_\mathcal{A} \mid \mathcal{D}_t)$, between the function output, $Y_\mathbf{x}$, and the algorithm output, $O_\mathcal{A}$. BAX assumes that functional output instances, $y_\mathbf{x}$, of the function, $f$, can be observed for each acquired $\mathbf{x}$. The acquisition of data is sequential, where the information gain maximization procedure leads to a dataset of observations, $\mathcal{D}_t := \{(\mathbf{x}_i, y_{\mathbf{x}_i})\}_{i=1}^{t-1}$, at step $t \in [T]$. BAX can be used in conjunction with a number of algorithms, such as determining the superlevel set (i.e. $\mathcal{X}_c$), computing integrals, or finding local optima of $f$. Given that genomic experimentation seeks to find a diverse set of genes corresponding to the modes of $f$, the BAX framework is well suited to our task.

Concretely, BAX acquisition functions select points by maximizing the expected information gain (EIG) obtained from each point about the output of the algorithm. Crucial to the applicability of BAX to our problem setting is the tractability of accurate approximators of the EIG for algorithms which, like the one we will propose, return a subset of their inputs. The exact computation of the EIG for arbitrary algorithms is not generally tractable; however, Neiswanger et al. (2021) present an approximation that only requires the computation of the entropy of the distribution over function values conditioned on algorithm outputs.

$$\text{EIG}_t^v(\mathbf{x}, \mathcal{D}_t) = H(f_{\text{ip}}(\mathbf{x})|\mathcal{D}_t) - \mathbb{E}_{p(S|\mathcal{D}_t)}[H(f_{\text{ip}}(\mathbf{x})|S, \mathcal{D}_t)]. \tag{1}$$

When the model $P$ is a Gaussian Process, both of these quantities are straightforward to compute: the first is the entropy of the GP's predictive distribution at $\mathbf{x}$, and we can estimate the second by conditioning a posterior on the values of elements in the set $S$. Monte Carlo approximation of this quantity is possible when the model $P$ does not permit a closed form.

## 3 PROBLEM SETTING

A primary challenge in the drug discovery pipeline is the discrepancy in outcomes between *in vitro* experimental data and *in vivo* diseases. Where *In vitro* experimental data can quantify the effect of a gene knockout on a specific aspect of a cellular phenotype in a petri dish, *in vivo* interactions between the drug and the organism may lead to weaker effect sizes or toxicity. The drug discovery pipeline consists of stages that start by testing a set of candidate interventions and then procedes by selecting a subset of promising candidates to pass on for further development. For example, one might test a broad range of gene knockouts on cell cultures and then select a subset to evaluate in animal models. These trials can be expensive, so it is desirable to weed out potentially ineffective or toxic candidates before this phase. To do so, researchers can leverage heuristic score functions that predict the "drug-like-ness" or likelihood of toxicity of a compound (Jiménez-Luna et al., 2020). Considering a diverse set of candidate interventions, where each intervention applies to a different mechanism in the disease phenotype, is also of use because it increases the likelihood of at least one candidate succeeding in the subsequent phase.

We formalize this problem as an optimization problem where the optimizer has access to a measurement correlated with the quantity of interest; however, it is noise augmented to emulate the primary objective function. We formalize our search space (i.e., the set of available genes, though in principle this could be any set) $\mathcal{G} = \{g_1, \ldots, g_m\}$, for which we have some phenotype measurement $f_{\text{ip}}$. We will primarily refer to $f_{\text{ip}}$ as a function from *features* to phenotype changes, but it is equivalent

to expressing $f_{\text{ip}}$ as a function on genes $\mathcal{G}$. The subscript 'ip' stands for *intermediate phenotype* as it is not the actual clinical measurement caused by the gene knockout. Instead, it is a measurement known to correlate with a disease pathology and is tractable in the lab setting (see Appendix B for detailed formalization). In this paper, we will assume the phenotype change is a real number $f_{\text{ip}}(\mathbf{x}) \in \mathbb{R}$; however, given suitable modeling assumptions, it is possible to extend our approach to vector-valued phenotype readouts. We also define a function called *disease outcome*, $f_{\text{out}}$, which is composed of $f_{\text{ip}}$ and factors outside the biological pathway, such as toxicity of a molecule that engages with a target gene. The noise component, $\eta$, encapsulates all these extra factors.

We consider two tractable formulations of the relationship between the disease outcome, $f_{\text{out}}$, and the *in vitro* phenotype, $f_{\text{ip}}$.

1. **Multiplicative Bernoulli noise:**

$$f_{\text{out}}(\mathbf{x}; \eta) = f_{\text{ip}}(\mathbf{x})\eta(\mathbf{x}) \tag{2}$$

where $\eta(\mathbf{x}) \in \{0, 1\}, \forall \mathbf{x} \in \mathcal{G}$, and $\eta$ is sampled from a Gaussian process classification model. This setting presents a simplified model of drug toxicity: $\eta$ corresponds to a binary indicator of whether or not the drug is revealed to exhibit unwanted side effects in future trials. The multiplicative noise model assumes that the downstream performance of an intervention is monotone with respect to its effect on the phenotype, conditional on the compound not exhibiting toxicity in future trials. In our experiments, we assume $\eta$ exhibits correlation structure over inputs corresponding to a GP classification model, and construct the kernel $K_{\mathcal{X}}$ of this GP to depend on some notion of distance in the embedding space $\mathcal{X}$.

2. **Additive Gaussian noise:**

$$f_{\text{out}}(\mathbf{x}; \eta) = f_{\text{ip}}(\mathbf{x}) + \eta(\mathbf{x}) \quad \eta \sim \text{GP}(\mathbf{0}, K_{\mathcal{X}}) \tag{3}$$

where $\eta : \mathcal{G} \to \mathbb{R}$ is drawn from a Gaussian process model with kernel $K_{\mathcal{X}}$. In this case, we assume that the unforeseen effects of the input $\mathbf{x}$ are sufficiently numerous to resemble a Gaussian perturbation of the measured in vitro phenotype $f_{\text{ip}}(\mathbf{x})$.

Notice that in the above models, noise is an umbrella term for everything that affects the fitness of a target but is not part of the biological pathway from the gene to the phenotype change. Therefore, the choice of noise distribution and how it affects the outcome is a modelling assumption that is intended to capture coarse inductive biases known to the researcher. We additionally seek out a *set* of interventions $S \subset \mathcal{G}$ of some fixed size $|S| = k$ whose elements cause the maximum expected change (for some noise distribution) in the disease outcome. In other words, we seek an intervention that best moves the disease phenotype, which will be the best candidate drug. This goal is distinct from either sampling the super-level-sets of $f_{\text{ip}}$ or finding the set $S$ with the best average performance. Instead, we explicitly seek to identify a set of points whose toxicity or unintended side effects will be minimally correlated, maximizing the odds that at least one will succeed in the subsequent trials. We thus obtain a set-valued maximization problem

$$\max_{S \subseteq \mathcal{X}} \mathbb{E}_{\eta} \left[ \max_{\mathbf{x} \in S} f_{\text{out}}(\mathbf{x}; \eta) \right] . \tag{4}$$

This compact formula is critical to attain our overarching objective: identifying interventions with both a large impact on the phenotype of interest and with high diversity to increase the chance of success of some of them in the subsequent steps of the drug discovery pipeline. An illustrative example is provided in Figure 6 in the Appendix to provide further intuition into this formula.

The general formulation of this problem is NP-hard (Goel et al., 2010); therefore, we propose a tractable algorithm that provides a constant-factor approximation of the optimal solution by leveraging the submodular structure of the objective under suitable modeling assumptions. Given such an algorithm, our task is the active learning problem of optimally querying the function, $f_{\text{ip}}$, given a limited number of trials, $T$, to accurately estimate the algorithm's output on the ground-truth dataset.

Importantly, this formulation allows us to decouple modeling the measured phenotype, $f_{\text{ip}}$, from modeling the noise $\eta$. For example, we might make the modeling assumption that we sample $f_{\text{ip}}$ from a GP with some kernel $k_1$ and that $\eta$ is a Bernoulli random variable indicating the safety of the compound.

# 4 METHOD

Various methods exist for efficiently optimizing black-box functions; however, our problem setting violates several assumptions underlying these approaches. In particular, while we assume access to intermediate readouts $f_{ip}$, the actual optimization target of interest $f_{out}$ is not observable. Further, we seek to find a *set* of interventions that maximize its expected value under some modeling assumptions. These two properties render a broad range of prior art inapplicable. Active sampling methods do not prioritize high-value regions of the input space. Bayesian optimization methods assume access to the ground-truth function outputs (or a noisy observation thereof). And Bayesian algorithm execution approaches based on level-set sampling may not sufficiently decorrelate the hidden noise in the outcome.

We propose an intervention set selection algorithm in a Bayesian algorithm execution procedure that leverages the modeling assumptions we characterize in the previous section. This method, Subset Discovery via Bayesian Algorithm Execution (DiscoBAX), consists of two distinct parts. (1) a subset-selection algorithm obtaining a $1 - 1/e$-factor approximation of the set that optimizes equation 3, and (2) an outer BAX loop that queries the phenotype readings to maximize the information gain about the output of this algorithm. In Section 4.1, we present the idealized form of DiscoBAX and show that it attains an approximately optimal solution. Our approach is easily adaptable to incorporate approximate posterior sampling methods, enabling its use with deep neural networks on high-dimensional datasets. We outline this practical implementation in Section 4.2.

## 4.1 ALGORITHM

**Subset maximization:** we first address the problem of identifying a subset $S \subset \mathcal{X}$ which maximizes the value $\mathbb{E}_\eta[\max_{\mathbf{x} \in S} f_{out}(\mathbf{x}; \eta)]$ As mentioned previously, the exact maximization of this objective is intractable. To construct a tractable approximation, we propose a submodular surrogate objective, under which the value of an intervention is lower-bounded by zero $f_{out}^*(\mathbf{x}; \eta) = \max(f_{out}(\mathbf{x}; \eta), 0)$. This choice is motivated by the intuition that any intervention with a negative expected value on the phenotype is equally useless as it will not be considered in later experiment iterations, and so we do not need to distinguish between harmful interventions. The resulting function $f(S) = \mathbb{E}_\eta[\max_{\mathbf{x} \in S} f_{out}^*(\mathbf{x}; \eta)]$ will be submodular, and thus Algorithm 1, the greedy algorithm, will provide a $1 - 1/e$ approximation of the optimal solution.

**Observation 1.** *The score function $f : \mathcal{P}(\mathcal{G}) \to \mathbb{R}$ defined by*

$$f(S) = \mathbb{E}_\eta \left[ \max_{\mathbf{x} \in S} \left( \max(0, f_{out}(\mathbf{x}; \eta)) \right) \right] \tag{5}$$

*is submodular.*

We provide proof of this result in Appendix A. In practice, we can estimate the expected value in this objective using Monte Carlo (MC) samples over the noise distribution $\eta$. Where MC sampling is too expensive, a heuristic that uses a threshold to remove points whose values under $\eta$ are too highly correlated can also obtain comparable results with a reduced computational burden.

---

**Algorithm 1** SubsetSelect (Multiplicative Noise)

**Require:** integer $k > 0$, set $\mathcal{X}$, distribution $P(\eta)$, sampled $\widehat{f}_{ip} : \mathcal{X} \to \mathbb{R}$
  $S \leftarrow \emptyset$
  $\widehat{f}_{out}(\mathbf{x}; \eta) := \widehat{f}_{ip}(\mathbf{x})\eta(\mathbf{x})$
  **for** $i < k$ **do**

    $S \leftarrow S \cup \{\arg\max_{x \in \mathcal{X} \setminus S} \mathbb{E}_\eta[\max_{y \in S \cup \{x\}} \widehat{f}_{out}(\mathbf{x}; \eta)]\}$

  **end for**
  **return** $S$

---

**Algorithm 2** DiscoBAX

**Require:** finite sample set $\mathcal{X}$, budget $T$, Monte Carlo parameter $\ell \in \mathbb{N}$
  $\mathcal{D} \leftarrow \emptyset$
  **for** $i < T$ **do**
    sample $\{\widehat{f}_{ip}\}_{j=1}^\ell \sim P(f_{ip}|\mathcal{D})$
    $S_j \leftarrow \text{SubsetSelect}(\widehat{f_{ip,j}}), \forall j = 1, \ldots, \ell$
    $\mathbf{x}_i \leftarrow \arg\max_{\mathbf{x} \in \mathcal{X}} \text{EIG}^v(\mathbf{x}, S_{j=1}^\ell)$
    query $f_{ip}(\mathbf{x}_i)$
    $\mathcal{D} = \mathcal{D} \cup \{(\mathbf{x}_i, f_{ip}(\mathbf{x}_i)\}$
  **end for**
  **return** $\mathcal{D}$

---

**Active sampling:** because we do not assume prior knowledge of the phenotype function $f_{ip}$, we require a means of selecting potential interventions for querying its value at a specified input $\mathbf{x}$. In practice, running these experiments may incur a cost, and so it is desirable to minimize the number of queries necessary to obtain an accurate estimate of the optimal intervention set. BAX (Neiswanger et al., 2021) presents an effective active sampling approach to approximate the output of an algorithm using a minimal number of queries to the dataset of interest. In our setting, this allows us to approximate the output of Algorithm 1 over the set $(\mathcal{X}, f_{ip}(\mathcal{X}))$ without incurring the cost of evaluating the effect of every knockout intervention in $\mathcal{G}$. Concretely, this procedure takes as input some probabilistic model $P$ which defines a distribution over phenotype readings $f_{ip}$ conditioned on the data $\mathcal{D}_t$ seen so far and from which it is possible to draw samples.

*A remark on the efficiency of subset maximization & active sampling—* It has to be emphasized that subset selection is a function called within each active sampling cycle. Hence, the above observation about submodularity refers specifically to Algorithm 1 rather than its incorporation in Algorithm 2. If sample efficiency is not a concern this algorithm could be run on the set of all inputs and provide the exact solution.

We outline this procedure in Algorithm 2, and refer to Section 2 for additional details. In the batch acquisition setting, we form batches of size $B$ at each cycle by selecting the $B$ points with the highest $EIG$ values.

### 4.2 PRACTICAL IMPLEMENTATION IN HIGH DIMENSIONS

When working with high-dimensional input features, we typically leverage Bayesian Neural Networks in lieu of Gaussian Processes. We sample from the parameter distribution via Monte Carlo dropout (MCD) (Gal & Ghahramani, 2016), and rely on Monte Carlo simulation to estimate the quantities introduced in Algorithm 2. In particular, the entropy of the posterior distribution is obtained as follows:

$$H(\mathbf{y_x}|\mathcal{D}_t) = \mathbb{E}_{p(\mathbf{y_x}|\mathcal{D}_t)}\left[\log p(\mathbf{y_x}|\mathcal{D}_t)\right] \sim \frac{1}{M}\sum_{s=1}^{M}\log p(y_x^s|\mathcal{D}_t, f_s) \tag{6}$$

where the samples $\{y_x^s = f_s(x)\}_{i=1}^{M}$ are obtained by sampling from the distribution over model parameters with MCD to obtain the parameter samples $\{f_s\}_{i=1}^{M}$.

## 5 EXPERIMENTS

In the experimental evaluation of DiscoBAX, we specifically seek to answer the following questions: 1) Does DiscoBAX allow us to reach a better trade-off between recovery of the top interventions and their diversity (Table 1 and 2)? 2) Is the method sample-efficient, i.e., identifies global optima in fewer experiments relative to random sampling or naive optimization baselines (Figure 3 and 5)? 3) Is the performance of DiscoBAX sensitive to various hyperparameter choices (Appendix D.3)? To address these questions, we first focus on experiments involving synthetic datasets (§ 5.1) in which we know the underlying ground truth objective function. We then conduct experiments across several large-scale experimental assays from the GeneDisco benchmark Mehrjou et al. (2021) that cover a diverse set of disease phenotypes.

### 5.1 SYNTHETIC DATA

We begin with a concrete example to illustrate the distinction between the behavior DiscoBAX and existing methods. The dataset we consider is a one-dimensional regression task on a mixture-of-Gaussians density function $f_{mog}$. We construct $f_{mog}$ such that it exhibits several local optima at a variety of values, necessitating a careful trade-off between exploration and exploitation to optimize the DiscoBAX objective. Crucially, exploitation in this setting requires not only an accurate estimation of the global optimum but also an accurate estimation of the local optima. We provide evaluations on additional datasets in Appendix D.1. We consider the following baseline acquisition functions which select the optimal point $\mathbf{x}^*$ to query at each iteration, letting $\mu(\mathbf{x})$ denote the posterior mean over $f_{ip}(\mathbf{x})$ and $\sigma^2(\mathbf{x})$ its variance. We evaluate random sampling, a UCB-like acquisition function, BAX on super-level set and top-k algorithms, Thompson sampling, and uncertainty maximization baselines. Full details are provided in Appendix D.1.

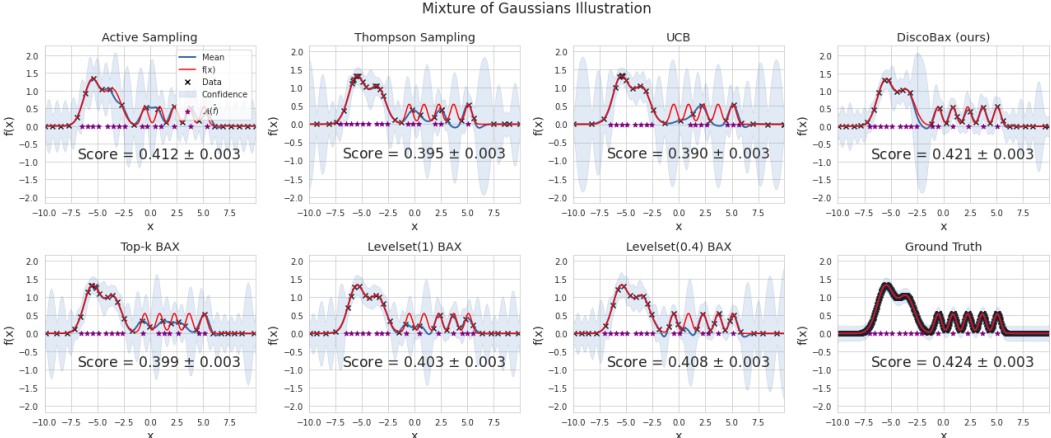

Figure 2: Illustration of failure modes of benchmark acquisition functions in our problem setting: existing methods struggle to accurately capture both the high- and low-valued local optima. We use a batch size equal to one for all methods.

In Figure 2, we visualize the solutions found by each approach after 30 iterations. We further evaluate the score of each method, computed as $\mathbb{E}_\eta \max_{\mathbf{x} \in S} f_{\text{ip}}(\mathbf{x})\eta(\mathbf{x})$, where $\eta$ is drawn from a Bernoulli distribution whose logits are determined by an affine transformation of a sample from a GP with zero mean and radial basis function covariance kernel. This construction ensures a high correlation between the values of nearby inputs and reward sets $S$ whose elements are distant from each other. To select $S$, we use the learned posterior mean $\mu$ from each acquisition strategy as input to Algorithm 1 and set $S$ to be equal to its output. We observe that most baselines over-exploit the high-value local optima, leading to inaccuracies on the lower optima. As a result, Algorithm 1 is unable to select the optimal subset elements from the lower-value modes and the model score suffers. The active sampling baseline yields a more uniform sampling distribution over inputs that results in a relatively uniform distribution of errors. While DiscoBAX does not perfectly estimate the value of the target function, its sampling strategy yields reasonably accurate estimates of all of the local optima.

## 5.2 GENEDISCO DATASET

**Datasets & baselines.** The GeneDisco benchmark (Mehrjou et al., 2021) is comprised of five large-scale genome-wide CRISPR assays and compares the relative strengths of nine active learning algorithms (eg., Margin sampling, Coreset) for optimal experimental design. The objective of the different methods is to select the set of interventions (ie., genetic knockouts) with the largest impact on the corresponding disease phenotype. We include all existing baselines from the GeneDisco benchmark, as well as eight additional approaches: UCB, qUCB, qEI, qPOI, Thompson sampling, Top-K BAX, Levelset BAX, and DiscoBAX.

**Metrics & approach.** We define the set of optimal interventions as the ones in the top percentile of the experimentally-measured phenotype (referred to as 'Top-K interventions'). We use the Top-K recall metric to assess the ability of the different methods to identify the best interventions. To quantify the diversity across the set of optimal interventions, we first cluster these interventions in a lower-dimensional subspace (details provided in Appendix C). We then measure the proportion of these clusters that are recalled (i.e., any of its members are selected) by a given algorithm over the different experiment cycles. The overall score of an approach is defined as the geometric mean between Top-K recall and the diversity metric. For all methods and datasets, we perform 25 consecutive batch acquisition cycles (with batch size 32). All experiments are repeated 10 times with different random seeds.

**Results & discussion.** We observe that, across the different datasets, DiscoBAX enables to identify a more diverse set of optimal interventions relative to baselines (Table 1). It does so in a sample-

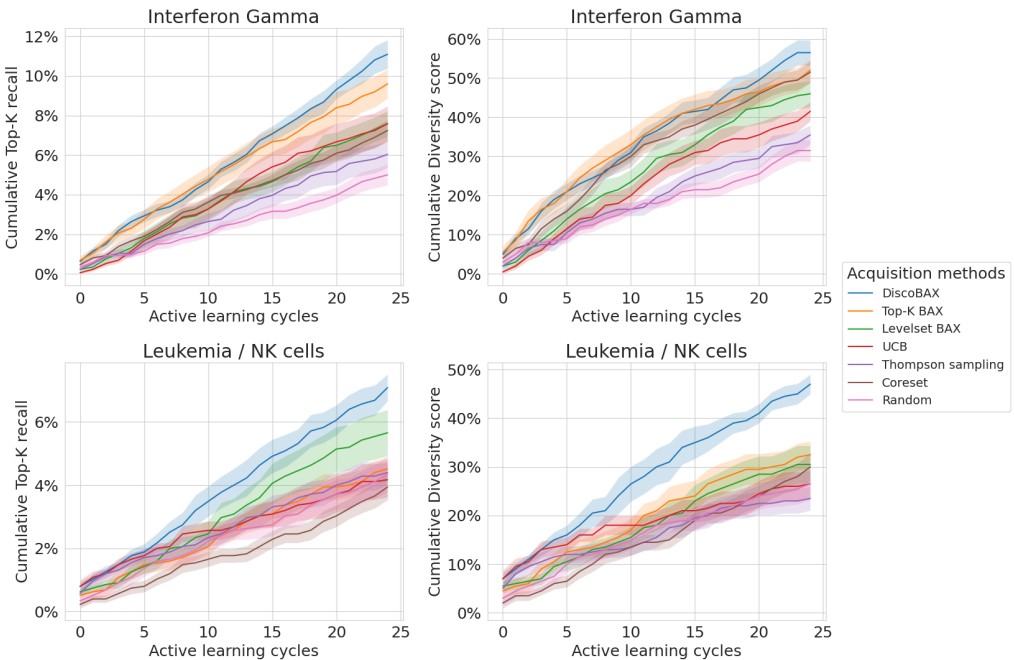

Figure 3: **Top-K recall and Diversity score Vs acquisition cycles** The two top plots are for the Interferon $\gamma$ assay (Schmidt et al., 2021), and the two bottom plots are based on the Leukemia assay (Zhuang et al., 2019).

efficient manner as it achieves higher diversity throughout the different acquisition cycles (Fig.3). Note that sample-efficiency is an empirical observation here not a theoretical property of the algorithm since it is possible to construct adversarial datasets where a BAX method will attain no better performance than random sampling. Interestingly, it tends to recall a higher share of optimal interventions on several assays as well, which may be the result of very steep extrema in the corresponding datasets. We also find the performance of DiscoBAX to be relatively insensitive to the choice of hyperparameters (Appendix D.3). Lastly, we note that when the input feature space (ie., the intervention representation) does not correlate much with the disease phenotype of interest, the model being learned tends to perform poorly and we observe no lift between the different methods and random sampling (eg., the SARS-CoV-2 assay from Zhu et al. (2021) – see Appendix D.2).

Table 1: **Performance comparison on GeneDisco CRISPR assays** We report the aggregated performance of DiscoBAX and other methods on all assays from the GeneDisco benchmark. All other baselines and the breakdown per assay are provided in Appendix D.2.

| Method | Category | Top-K recall | Diversity score | Overall score |
|---|---|---|---|---|
| Random | - | 29.3% (1.4%) | 4.9% (0.3%) | 12.0% (0.6%) |
| Thompson Sampling | Bandits | 27.5% (1.5%) | 4.8% (0.4%) | 11.5% (0.7%) |
| UCB | Bayesian Optim. | 33.5% (2.0%) | 5.9% (0.5%) | 14.1% (1.0%) |
| Coreset | Active learning | 39.3% (1.9%) | 5.5% (0.3%) | 14.7% (0.8%) |
| Levelset Bax | BAX | 35.4% (2.2%) | 6.3% (0.4%) | 15.0% (0.9%) |
| Top-K Bax | BAX | 38.8% (2.3%) | 6.8% (0.6%) | 16.2% (1.2%) |
| DiscoBax (ours) | BAX | **44.1% (2.2%)** | **7.8% (0.5%)** | **18.6% (1.1%)** |

## 6 RELATED WORK

Prior works have studied the application of genomic discovery and method development for diverse target generation.

**Bayesian optimization:** Bayesian optimization (BO) is concerned with finding the global optimum of a function with the fewest number of function evaluations (Snoek et al., 2012; Shahriari et al., 2015). Since this target function is often expensive-to-evaluate, one typically uses a Gaussian process as a surrogate function (Srinivas et al.). The candidates for function evaluation are then determined through a so-called acquisition function, which is often expressed as the expected utility over the surrogate model. Typical choices include the expected improvement (Močkus, 1975, EI) and probability of improvement (Kushner, 1964, PI) as utility functions. Recent work includes variational approaches Song et al. (2022) which yield a tractable acquisition function whose limiting behavior is equivalent to PI. Prior work tried to obtain diversity in Bayesian optimization e.g. through a batch setting (Kirsch et al., 2019) or multi-objective optimization (Hernández-Lobato et al., 2016). Bayesian optimization has been applied to biological problem settings such as small molecule optimization (Korovina et al., 2020) or automatic chemical design (Griffiths & Hernández-Lobato, 2017).

**Optimal experiment design** broadens the scope of Bayesian Optimization: rather than simply maximizing a parametric function, the task is to adaptively identify an optimal set of experiments to efficiently reach some goal (Robbins, 1952; Chernoff, 1959). Applying machine learning to automate hypothesis generation and testing goes back multiple decades (King et al., 2004). Optimal experiment design is amenable to Bayesian optimization (Greenhill et al., 2020) and reinforcement learning approaches (Kandasamy et al., 2019). Most related to our work is Bayesian Algorithm Execution (BAX) Neiswanger et al. (2021) that extends the goal of experiment design from only finding the maximum of a function to estimating more general properties such as level sets by computing the expected information gain (EIG) which is the mutual information between the evaluation of an input point and the statistics related that property.

**Active learning** While many probabilistic models like Gaussian processes provide principled uncertainty estimates (Rasmussen, 2003), modern neural network architectures often rely on heuristics or only provide approximations approaches (Gal & Ghahramani, 2016; Lakshminarayanan et al., 2017). Active learning based approaches use the uncertainty estimates for maximizing expected information gains of model parameters (Houlsby et al., 2011). Recently, more and more approaches have used active learning based on model uncertainties of neural networks for biomedical applications.

**Bandits:** The upper confidence bounds seen in BO originate in the bandit setting (Lai & Robbins, 1985), in which one can extend the widely-used UCB algorithm to Gaussian processes (Grünewälder et al., 2010; Srinivas et al.). While both bandits and BO seek to find the maximum of a function, the two problem settings leverage different notions of optimality. BO seeks to *identify* the argmax, whereas bandits seek to *minimize* the number of sub-optimal queries. Related to bandits and BO, some efforts are made to formulate active learning as a reinforcement learning problem (Slade & Branson, 2022; Casanova et al., 2020; Konyushkova et al., 2017; Pang et al., 2018).

## 7 CONCLUSION

We have introduced a mathematical formalization of the drug discovery problem that captures the noise induced by moving from in vitro to in vivo experiments. We proposed a novel algorithm based on Bayesian Algorithm Execution and illustrated its utility on many illustrative synthetic datasets. We have further evaluated this class of methods against the real-world large-scale assays from the GeneDisco benchmark, where they help identify diverse top interventions better than existing baselines. Future work could see the extension of the current framework to explicitly account for the fact that experimental cycles happen in batches. Further, we assume in this work that distant representations of interventions implied different underlying biological mechanisms - a proper causal formulation of the problem would allow us to tell apart causally connected pathways more cleanly. Finally, it is typical practice to measure several potential intermediate phenotypes of interest to capture different aspects of interest, which requires an extension of our approach to the setting of multiple objectives.

## 8 REPRODUCIBILITY STATEMENT

We clearly state our modelling assumptions throughout Sections 2 to 4. We provide proof for our theoretical claims in Appendix A. All experimental results reported in Section 5 and appendix D can be reproduced using the code available at: `https://github.com/anonymous35780/solaris-2023-iclr`. Hyper-parameter sweeps for the BAX methods for GeneDisco are presented in Table 3.

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
