# OpenReview forum: "DiscoBAX - Discovery of optimal intervention sets in genomic experiment design"
_ICLR.cc/2023/Conference — Submitted to ICLR 2023_

### Official Review · Reviewer_uzVy · 2022-10-22

**Confidence:** 4
**Correctness:** 3
**Technical Novelty And Significance:** 2
**Empirical Novelty And Significance:** 3
**Recommendation:** 6

**Clarity, Quality, Novelty And Reproducibility:**

The problem setting is clearly defined and the usage of BAX style approach is novel. It is commendable that source code is made available for easy reproducibility.

**Strength And Weaknesses:**

- The problem considered in the paper is important with real-world implications. I found the biological motivation of the problem well-written with good intuitions for a non-domain person.

- The overall idea of using BAX style approach for this problem setting is fairly interesting and novel.

However, I have few questions  to understand some of the details better and some suggestions that will hopefully improve the paper's contributions:

- It is mentioned in the paper that the discrepancy between the disease outcome $f_{out}$ and intermediate phenotype $f_{ip}$ is captured by the noise distribution $\eta$. It is motivated as an important distinguishing factor between the problem setting of this paper compared to that of existing approaches like Bayesian optimization. However, there is little clear description about estimating this quantity or principles behind choosing a certain distribution. A short remark is mentioned after observation 1 in few lines but that seems limited for such an important motivation of the problem setting. Please describe the concrete implementation/algorithmic details of the noise distribution and how it is estimated in the experiments.

- The proof of submodularity of the score function in appendix A is missing some description. For example, $dis(g, \eta)$ is not defined. Please expand the proof and explain the reasoning behind the inequalities. If it is `straightforward` (as mentioned in the first line of the proof), can it be considered a major contribution of the paper?

- The score function is chosen to be the best value in a set (averaged over noise distribution eta). This seems like an optimistic choice. For the algorithm to find good robust points, should we not consider the worst value in the set as the score function?

- Please add a description of the computational complexity of the proposed approach. Since multiple instances of the subset maximization algorithm needs to be run in each iteration for estimating the expected information gain quantity, the computational complexity can be quite high. A wall-clock time comparison of the proposed approach with existing baselines will be very useful, especially since the top-k recall gap is relatively small on most of the benchmarks (other than Leukemia/NK cells).

- As mentioned in section 5.2, batch size of 32 points are selected for evaluation in each acquisition cycle. How is the batch of points selected? Is it accomplished by greedy optimization of the EIG acquisition function? If yes, what are the accuracy losses of doing this greedy optimization.

- The choice of UCB as Bayesian Optimization (BO) baseline strategy seems surprising. There is a large literature on Bayesian optimization algorithms (please see [1-4] and references therein) for the batch evaluation setting that should be the right comparison here. Something like qEI (q-Expected Improvement) is easy to setup and implemented in popular packages like BoTorch [1] (https://botorch.org/tutorials/). The points suggested by qEI are also known to find highly diverse points in the context of batch optimization. Please consider improving this comparison (by including qEI for instance) because BO algorithms are directly applicable and the most relevant baseline in the setting.

References


[1] Balandat, Maximilian, Brian Karrer, Daniel Jiang, Samuel Daulton, Ben Letham, Andrew G. Wilson, and Eytan Bakshy. "BoTorch: a framework for efficient Monte-Carlo Bayesian optimization." Advances in neural information processing systems 33 (2020): 21524-21538.
González, J., Dai, Z., Hennig, P., & Lawrence, N. (2016, May). Batch Bayesian optimization via local penalization. In Artificial intelligence and statistics (pp. 648-657). PMLR.

[2] Wu, J., & Frazier, P. (2016). The parallel knowledge gradient method for batch Bayesian optimization. Advances in neural information processing systems, 29.

[3] Azimi, J., Fern, A., & Fern, X. (2010). Batch bayesian optimization via simulation matching. Advances in Neural Information Processing Systems, 23.

[4] Gong, C., Peng, J., & Liu, Q. (2019, May). Quantile stein variational gradient descent for batch Bayesian optimization. In International Conference on Machine Learning (pp. 2347-2356). PMLR.



**Summary Of The Paper:**

The paper considers the problem of finding a `diverse` set of genomic interventions that maximize a phenotype of interest. Formally, the problem can be modeled as optimizing expensive black-box functions over sets of inputs from a given design space. Bayesian Algorithm eXecution (BAX) is a recently proposed framework that allows estimating required properties of black-box functions using information gain based input acquisition strategies. The paper provides a BAX (Bayesian `Algorithm` eXecution) style approach to solve the problem where the key idea is to consider the subset maximization of a chosen score function as the `Algorithm`. At each iteration, an Expected Information Gain acquisition objective is optimized to select the next points for evaluation. This objective is parameterized by multiple sample outputs of the subset maximization algorithm. Experiments are performed on synthetic and GeneDisco benchmarks.

**Summary Of The Review:**

I find the overall approach to be quite interesting while addressing an important problem. However, some of the design choices and baseline comparisons require more work to improve the quality of the paper for an ICLR publication.

---

> ### Author Response · Authors · 2022-11-19
> **Author Response to Official Review by Reviewer uzVy (Part 1)**
>
> Thank you for the very thoughtful feedback and recognizing our work as a novel contribution with real-world implications. We address each of your concerns below. Changes to the text (eg., clarifications, new experiments) are coloured red in the revised manuscript.
> (Cx: Reviewer’s Concern number x, AR: Author Response)
>
> C1: **It is mentioned in the paper that the discrepancy between the disease outcome and intermediate phenotype is captured by the noise distribution. It is motivated as an important distinguishing factor between the problem setting of this paper compared to that of existing approaches like Bayesian optimization. However, there is little clear description about estimating this quantity or principles behind choosing a certain distribution. A short remark is mentioned after observation 1 in few lines but that seems limited for such an important motivation of the problem setting. Please describe the concrete implementation/algorithmic details of the noise distribution and how it is estimated in the experiments.**
>
> AR: The noise distribution in the model studied by this paper is intended to capture the aspects of drug interaction in future trials over which the experimenter has some uncertainty. The choice of noise distribution is thus a modeling assumption, intended to capture coarse inductive biases known to the researcher, for example that interventions on genes which affect similar biological pathways are likely to have similar side effects. The objective of our method is to propose a set of drug candidates which, based on a set of modeling assumptions on their correlation structure, maximize the odds of at least one succeeding in further trials by decorrelating this risk of side effects. While we agree that in an ideal world these modeling assumptions could be empirically validated, in practice this may not be tractable to the expensive and unpredictable nature of biomedical experiments. Our problem formulation therefore assumes that the experimenter does not have access to this information, but may still have reasonable inductive biases which will enable better performance than random guessing. In the revised manuscript, we emphasized this point in the paragraph before Eq. (4) and after introducing the two noise models.
>
> C2: **The proof of submodularity of the score function in appendix A is missing some description. For example, dis is not defined. Please expand the proof and explain the reasoning behind the inequalities. If it is straightforward (as mentioned in the first line of the proof), can it be considered a major contribution of the paper?**
>
> AR: We apologize for the confusion in the proof in appendix A. This was due to an incorrectly formatted macro. “Dis” in this derivation is simply $f_{\mathrm{out}}$. The proof should be straightforward with this clarification, and we have corrected the error in our revisions.
>
> Regarding the contribution of the paper, it is not the _proof_ of the submodularity of the score function which constitutes the scope of the contribution, but the construction of the method which uses it and the formalization of the problem it addresses. The acquisition function we propose is both tractable for approximate optimization and captures many of the properties we outline as desirable for drug discovery, and it is these properties which constitute the novelty and significance of the method.
>
> C3: **The score function is chosen to be the best value in a set (averaged over noise distribution eta). This seems like an optimistic choice. For the algorithm to find good robust points, should we not consider the worst value in the set as the score function?**
>
>
> AR: Our choice of score function is motivated by the realities of the drug discovery process, which is best viewed as a search problem. The success of a drug discovery pipeline is determined by the effect of the best candidate found by the process on the target disease phenotype, as this will be the candidate pushed forward as a drug. If a set of compounds are evaluated and only one demonstrates a benefit in clinical trials, then the set of candidate drugs will be considered a success independent of the effects of the failed compounds.

---

> > ### Author Response · Authors · 2022-11-19
> > **Author Response to Official Review by Reviewer uzVy (Part 2)**
> >
> > C4: **Please add a description of the computational complexity of the proposed approach. Since multiple instances of the subset maximization algorithm needs to be run in each iteration for estimating the expected information gain quantity, the computational complexity can be quite high. A wall-clock time comparison of the proposed approach with existing baselines will be very useful, especially since the top-k recall gap is relatively small on most of the benchmarks (other than Leukemia/NK cells).**
> >
> > AR: The main burden of computation in our method comes from the Monte Carlo sampling needed to estimate the information gain quantity in Algorithm 2 and the expectation over random noise in Algorithm 3; the overall complexity will be linear in the budget used to estimate each of these quantities. While this can present some computational overhead, we have found that the practical implementation of our approach on the GeneDisco dataset is on the order of a few hours. This is dwarfed in practice by the time and resources required to conduct the corresponding biological experiments at each cycle and is therefore not a practical concern. We also emphasize that a naive implementation of the exact maximization of Eq. (4) would have time exponential in the size of the candidate set $S$, which would become prohibitive for sets larger than a few dozen interventions; our greedy algorithm in contrast runs in time linear in $|S|$.
> >
> > C5: **As mentioned in section 5.2, batch size of 32 points are selected for evaluation in each acquisition cycle. How is the batch of points selected? Is it accomplished by greedy optimization of the EIG acquisition function? If yes, what are the accuracy losses of doing this greedy optimization.**
> >
> > AR: That is correct: to form a batch of size B, we select the B points with the highest EIG acquisition function values. We clarified that point clearer in the revised manuscript in section 4.1. We do agree with the reviewer that this is an approximation and that further improvements could be achieved there. As briefly mentioned in the conclusion, adapting the framework to account for dependencies between points in a batch setting is a research avenue for future work.
> >
> >
> > C6: **The choice of UCB as Bayesian Optimization (BO) baseline strategy seems surprising. There is a large literature on Bayesian optimization algorithms (please see [1-4] and references therein) for the batch evaluation setting that should be the right comparison here. Something like qEI (q-Expected Improvement) is easy to setup and implemented in popular packages like BoTorch [1] (https://botorch.org/tutorials/). The points suggested by qEI are also known to find highly diverse points in the context of batch optimization. Please consider improving this comparison (by including qEI for instance) because BO algorithms are directly applicable and the most relevant baseline in the setting.**
> >
> >
> > AR: Thank you for the suggestion. We had chosen UCB as it is a commonly-used Bayesian Optimization (BO) method, but agree with the reviewer that other BO methods adapted to the batch setting are very relevant here. In the synthetic dataset experiments where the batch size used was equal to 1, we added EI as a new baseline (please see the newly added section F in Appendix and Figure 6). In the GeneDisco experiments, we have carried out new analyses with three new baselines (qEI, qUCB and qPOI -- where POI stands for Probability of Improvement), all implemented with the BoTorch package. We observe improved results over the UCB baseline, yet the new baselines do not bridge the gap with the BAX methods and DiscoBAX in particular.

---

### Official Review · Reviewer_JiqB · 2022-10-23

**Confidence:** 3
**Correctness:** 3
**Technical Novelty And Significance:** 2
**Empirical Novelty And Significance:** 3
**Recommendation:** 5

**Clarity, Quality, Novelty And Reproducibility:**

The paper does not make its contributions clear, and I do think that it is currently an issue. Otherwise, the paper reads well. The application of BAX to genomics is novel (but the practical implications for the field of genetics are rather unclear), but the improvement over BAX is not clear (as the paper is written now).

**Strength And Weaknesses:**

Strength:
(1) Novelty *in its application*: this is one of the (relatively) few papers that tackle this important scientific problem (other example I could find is https://arxiv.org/pdf/2207.12805.pdf).
(2) The paper shows improved results in the recent GeneDisco benchmark, for a few of the datasets.

Weaknesses:
(1) My main issue is that I am having a hard time understanding the *technical* novelty of the paper. A lot of the concepts explained in Section 4 seems to belong more to a background section, as they are not contributions of this particular work, but from the BAX paper. As far as I understand, the contributions are (a) working with sets, and performing greedy set optimization and (b) using BNN for uncertainty quantification. However, in the results section, I don't see a clear explanation that those changes are improving the performance. It seems that (a) improves diversity? (if so, it should be made clear), but (b) has no comparison to different flavors of BNNs, GPs etc..
(2) there is no theoretical analysis on how the (1 - 1/e) approximation error propagates through the BAX procedure. Then, it is not correct for the author to write "theoretical guarantees on the optimality of the approach". Similarly, the authors write that DiscoBAX is "sample-efficient", but I have seen no proofs of sample efficiency, and the experimental results are not necessarily strong enough to claim this.
(3) There is some improvement over BAX, but only in a few datasets (2 / 4).
(4) There should be a comparison to other BNN flavors, and at the very least a discussion of uncertainty modeling w/ neural networks


**Summary Of The Paper:**

This paper is concerned with the iterative selection of an optimal set of targets for genetic interventions based on a scalar readout. The methods section described a particular instance of BAX, that models the singularity of the biological problem that is considered. A few aspects that makes it different from the existing InfoBAX approach is (1) the subset maximization target function and (2) a practical uncertainty model with BNN instead of GPs. Then, the paper provides a set of experiments on synthetic data, and on the openly available GeneDisco benchmark.

**Summary Of The Review:**

This paper provides an interesting application of BAX, and investigate a practical variant to batch learning of genetic perturbations. Beyond this, the technical contributions may be considered weak, and the empirical results are not exactly strong either (please refer to my previous sections for actionable items). Regarding the experimental results, I completely understand that it might be caused by noise in some datasets (e.g., COVID). In this case, I do think that the authors should carefully present ablation studies so that readers can understand what features of DiscoBAX helps in increase performance compared to BAX (again, see above for suggestions). In the light of this, I am currently leaning towards rejection of this manuscript, but very much looking forward to discussions with authors, and other reviewers.

---

> ### Author Response · Authors · 2022-11-19
> **Author Response to Official Review by Reviewer JiqB (Part 1)**
>
> We thank the reviewer for their helpful feedback and for recognizing the novelty of this work in the important area that it investigates, i.e. the intersection of genetics and machine learning. We address each of the reviewer’s concerns in detail below. Overall, we find that the reviewer had perhaps under-appreciated the technical contributions from our work (please read in particular responses to C1 and C2, as well as responses to C3 and C5 from reviewer JiqB) and we further clarified the strengths of empirical results (in C3 below and in the revised manuscript). We thus kindly ask the reviewer to reconsider their scoring for the technical novelty and the main recommendation.
> (Cx: Reviewer’s Concern number x, AR: Author Response)
>
> C1: **My main issue is that I am having a hard time understanding the technical novelty of the paper. A lot of the concepts explained in Section 4 seem to belong more to a background section, as they are not contributions of this particular work, but from the BAX paper. As far as I understand, the contributions are (a) working with sets, and performing greedy set optimization and (b) using BNN for uncertainty quantification. However, in the results section, I don't see a clear explanation that those changes are improving the performance. It seems that (a) improves diversity? (if so, it should be made clear), but (b) has no comparison to different flavors of BNNs, GPs etc.**
>
> AR: We agree with the reviewer that Section 4 describes a number of details of the BAX algorithm that, while necessary to understand how our contribution fits into that framework, are indeed prior work. We have moved these elements to Section 2 in order to better highlight the novel contributions of the paper, which are as follows: 1) formalizing the drug discovery problem as a maximization over a set of random variables in Eq. (4); 2) constructing a greedy optimization algorithm which can obtain a constant-factor approximation of the optimal solution to this objective; 3) demonstrating empirically that practical implementations of this algorithm can succeed at identifying promising sets of candidate interventions by leveraging the BAX framework to improve sample efficiency and BNNs to obtain uncertainty estimates over high-dimensional datasets.
> We emphasize that the maximization of the expectation of the maximum of a set of correlated random variables in Eq. (4) means that methods which seek to identify only the maximum over observable phenotype changes will achieve suboptimal performance on this objective. Thus contribution (2) differs significantly from prior work, which does not explicitly seek to decorrelate risk between set members. The improved diversity of candidate points achieved by our method is illustrated in both the synthetic and GeneDisco experiments: in the synthetic experiments, this is demonstrated by the success of DiscoBAX at identifying all modes of the distribution; in the GeneDisco experiments, this is demonstrated by the higher recall rate and diversity metric (Table 1).
>
> C2: **There is no theoretical analysis on how the (1 - 1/e) approximation error propagates through the BAX procedure. Then, it is not correct for the author to write "theoretical guarantees on the optimality of the approach". Similarly, the authors write that DiscoBAX is "sample-efficient", but I have seen no proofs of sample efficiency, and the experimental results are not necessarily strong enough to claim this.**
>
> AR:We note first that Algorithm 1 does indeed provide a constant-factor approximation of the optimal solution with respect to the input function, and that if sample efficiency is not a concern this algorithm could be run on the set of all inputs and provide the exact solution. We clarified this point in Section 4.1 of the revised manuscript. Due to the dependence of BAX on the fit between the prior distribution over functions and the real-world data, it is possible to construct adversarial datasets where a BAX method will attain no better performance than random sampling. While it is theoretically possible to construct such examples, we note that in practice BAX has demonstrated impressive empirical results, which motivates our discussion of sample efficiency. We have clarified in the Results & discussion paragraph in Section 5.2 that we are referring to the empirically observed sample efficiency of BAX, rather than to a theoretical property of the method.

---

> > ### Author Response · Authors · 2022-11-19
> > **Author Response to Official Review by Reviewer JiqB (Part 2)**
> >
> > C3: **There is some improvement over BAX, but only in a few datasets (2 / 4).**
> >
> > AR:  As a first point of clarification, BAX is not a separate method that we compare against but rather an overarching framework that encompasses several methods, including ours. DiscoBAX outperforms all other 13 baselines (16 in the revised manuscript) we compare against (BAX-based and others) on 3 out of the 5 datasets included in the GeneDisco benchmark (see Table 3 in Appendix), performs on par (significant overlap of confidence intervals) with the best methods on the 4th one (Tau protein) and is only outperformed by “random selection” on the last one (SARS-coV2). As discussed in section 5.2 and as noted in [1], the fact that random outperforms all other 16 methods on that dataset seems to indicate an issue with the data (eg., the feature space does not correlate with the disease phenotype, high label noise), rather than an algorithmic issue. Critically, none of the other baselines performs consistently high on all 5 assays: for instance, ‘random’ performs relatively poorly on all other 4 assays and the other methods that are on par with DiscoBAX on the Tau protein assay (eg., BADGE, Coreset) have inconsistent performance on other assays. Based on the feedback from the reviewer, we understand that the presentation of results was not conveying this message very clearly. To that end, we have updated Table 1 in the main text to reflect average performance across all 5 datasets, keeping the details of the per dataset analysis to the Appendix. We have also added the clarifications above to the corresponding section in Appendix (section D.2).
> >
> >
> > C4: **There should be a comparison to other BNN flavors, and at the very least a discussion of uncertainty modeling w/ neural networks**
> >
> >
> > AR: We would like to emphasize that the main contribution of this paper is the acquisition function (experiment designer) and in this sense, our contribution is agnostic to the choice of model used to generate the Bayesian posterior, provided this model provides sufficiently accurate samples. Depending on the problem constraints and the structure of the input space, other methods may be preferable over BNNs. For example, [1] show that, in some settings, ensemble methods (e.g. random forests) outperform other approaches. Because the models we use seem to be a reasonably good fit for the data (as most acquisition methods using these models outperform random on most datasets), we did not expect that further evaluations of additional uncertainty models would provide much additional insight into our acquisition function. If the reviewer still disagrees, we would be happy to discuss further or run additional experiments, given clarification on what is intended by “other flavors” of BNN.
> >
> >
> > [1] Mehrjou, A., Soleymani, A., Jesson, A., Notin, P., Gal, Y., Bauer, S., & Schwab, P. (2022). GeneDisco: A Benchmark for Experimental Design in Drug Discovery. ICLR 2022.

---

### Official Review · Reviewer_efcP · 2022-11-09

**Confidence:** 4
**Correctness:** 2
**Technical Novelty And Significance:** 2
**Empirical Novelty And Significance:** 2
**Recommendation:** 3

**Clarity, Quality, Novelty And Reproducibility:**

I have to give the authors top marks for reproducibility since the code is available.

A problem with the paper is that it is hard to me to assess novelty. Although the authors do lay out their contributions at the beginning, it's not clear to me whether this paper or another is the source of the technical inventions. For example, the submodularity of the relu'd score function.

A minor (clarity) point, but the figures are poor quality. I cannot read the axis labels when printed out. Unfortunately this gives the impression of a rushed paper.

**Strength And Weaknesses:**

Strengths
--
After spending some time with the paper, I came to appreciate the formalisation of the setting that is being worked in. I particularly enjoyed section 3 which laid out the problem. I appreciate the authors use of the appendices to keep long proofs out of the paper (the submodular proof for example) and also the clear presentation of the algorithms used. Posting the code anonymously gives me some confidence that the work is sound.

Weaknesses
--
the paper fails to deliver what is promised at the start of the experimental section - an investigation of discoBAX's sample efficiency, sensitivity to hyper-parameter settings, and comparison with the top-intervention method. As a result I feel that the paper is really rather unfinished! I am keen to understand more about discoBAX and am disappointed that the investigation is not deeper.

The 'related work' section at the back does not cover BAX - which appear to be prior art? this section feels like filler, I would have preferred the space be used to investigate discoBAX further.

I did not find a satisfactory discussion (or empirical evaluation) of the use of the relu'd score function, relative to the original objective. I get that this makes the optimization submodular, but have we lost anything? In which cases does this cause a problem?

It took me some time to work out what was going on - I think more discussion around Figure 1 is needed, and perhaps some commentary on how this is connected to the invitro/in-vivo setting. I liked the clarity of eq 3, but this felt like a long time coming, and would have been better placed at the start of the paper perhaps?


**Summary Of The Paper:**

The paper considers the problem of designing an experiment where there are two stages: in the first (in vitro) stage our task is to (efficiently) design an experiment that will have good chance of success in the second (in vivo) stage. The authors formalize this elegantly in equation (3).

The proposed solution is discoBAX: an algorithm that actively searches the in-vitro function for a set of optimal yet diverse points that have the best shot at having a successful outcome in the in vivo stage.

The paper explores discoBAX on a synthetic and real-world dataset.



**Summary Of The Review:**

A really interesting setting to be working in, and what seems like a good idea, but a lack of empirical investigation into the method lets the paper down. Clarity is a problem and it's difficult to assess which parts of the work are a novelty. I could change my "technical novelty' score if the authors could clarify.

---

> ### Author Response · Authors · 2022-11-19
> **Author Response to Official Review by Reviewer efcP (Part 1)**
>
> We thank the reviewer for their thoughtful review and suggestions for improvements. We provide detailed responses to all points of feedback below, but summarize the key points here:
> 1. We further clarified how the objectives set forth at the beginning of the experimental section (section 5) are addressed in the original manuscript (see response to C1)
> 2. We revised the manuscript to address the reviewer’s suggestions as it pertains to technical novelty (see responses to C3 and C5 below, as well as response C1 to reviewer JiqB)
> 3. We provided further intuition to the reader into the inner workings of our proposed acquisition function (responses to C4)
> 4. We fixed all other minor points of feedback as suggested (responses to C2 and C6)
>
> Changes to the text (eg., clarifications, new experiments) are coloured red in the revised manuscript.
>
> (Cx: Reviewer’s Concern number x, AR: Author Response)
>
> C1: **The paper fails to deliver what is promised at the start of the experimental section - an investigation of discoBAX's sample efficiency, sensitivity to hyper-parameter settings, and comparison with the top-intervention method. As a result I feel that the paper is really rather unfinished! I am keen to understand more about discoBAX and am disappointed that the investigation is not deeper.**
>
> AR: The experimental section specifically covers these different points as follows:
> 1. The sample efficiency is demonstrated both by the higher recall rate and diversity score for DiscoBAX relative to baselines *throughout the different active learning cycles* (Figure 3)
> 2. The robustness to hyperparameters relative to baselines is discussed in the text (Section 5.2) with a reference to the detailed analysis in the Appendix (Section D.3 and Table 3)
> 3. We compare with *thirteen* baselines (*sixteen* in the revised manuscript) from the relevant literature and demonstrate superior performance across experiments (Table 1)
> We also designed the experiments in Section 5.1 to provide intuition to the reader about the inner workings of the approach and how it compares with existing baselines.
>
> This seems to be your primary point of feedback in the review. As detailed above, we do believe we hit on all these points in the original manuscript and hope the above clarifications will alleviate your concern (we also adjusted the beginning of section 5 with these clarifications in the revision). Please let us know if that is not the case. We also found this particular point of feedback to be a bit underspecified / not actionable written as such.  Thus, should concerns remain, could you please let us know if there were other specific experiments that you were expecting to see in this section but were not included?
>
> C2: **The 'related work' section at the back does not cover BAX - which appear to be prior art? this section feels like filler, I would have preferred the space be used to investigate discoBAX further.**
>
>
> AR: Since our method belongs to the overarching BAX framework, we described the key characteristics of BAX needed for understanding our work in section 2 (Background and notation). While we had not included BAX in the related works sections to avoid repetition, we do hear the reviewers’ point and thus added a brief reference to it for comprehensiveness in section 6 (Related work). However, we do believe that this section is both critical to understand how our work connects to different yet related literature, as well as providing context for the different baselines leveraged in experiments (section 5). Readers who are perhaps a bit less knowledgeable about the space than the reviewer will likely find it helpful and, given the interdisciplinary nature of our work, we do want to cater for these readers as well.

---

> > ### Author Response · Authors · 2022-11-19
> > **Author Response to Official Review by Reviewer efcP (Part 2)**
> >
> > C3: **I did not find a satisfactory discussion (or empirical evaluation) of the use of the relu'd score function, relative to the original objective. I get that this makes the optimization submodular, but have we lost anything? In which cases does this cause a problem?**
> >
> > AR: The justification for the floor on the phenotype effect stems from the nature of the problem setting we consider, which can be analogized to best-arm identification in a bandit where one arm is equivalent to doing nothing, and the remainder constitute candidate interventions. In particular, if no candidates in a set move the phenotype in the desired direction, then the utility we have gained from this set is equal to the best intervention (i.e. the empty intervention) which has value zero, motivating the use of the relu’d score function. This has the added benefit of making the score function submodular, but is grounded in the nature of search problems.
> >
> >
> > **C4: It took me some time to work out what was going on - I think more discussion around Figure 1 is needed, and perhaps some commentary on how this is connected to the invitro/in-vivo setting. I liked the clarity of eq 3, but this felt like a long time coming, and would have been better placed at the start of the paper perhaps?**
> >
> > AR: Figure 1 illustrates our overarching objective which is to identify interventions with both a large impact on the phenotype of interest and with high diversity. The connection with genomic experiments and the in-vitro/in-vivo setting is discussed in the first paragraph of introduction: the diversity of interventions is desirable to increase the chance of success of some of them in the subsequent steps of the drug discovery pipeline (“if a promising candidate based on in-vitro experiments triggers unexpected issues when tested in-vivo (e.g., undesirable side effects), other lead precursors relying on different pathways might be suitable replacements that are not subject to the same issue”). Based on your suggestion, we wrote an extensive discussion for this figure in the Appendix. E of the revised manuscript. We also made that last connection to the in-vitro/in-vivo setting clearer in the caption of Figure 1.
> >
> > We agree with the reviewer that Equation (3) (or Eq. (4) in the revised manuscript) succinctly captures key aspects of our proposed solution. However it does require relevant background and motivations to be properly introduced. Based on the reviewer’s suggestion though, we included an additional sentence in the Introduction with a pointer to that equation to allow readers who are already comfortable with the background to jump ahead more rapidly to the important part of our solution.
> >
> > **C5: A problem with the paper is that it is hard for me to assess novelty. Although the authors do lay out their contributions at the beginning, it's not clear to me whether this paper or another is the source of the technical inventions. For example, the submodularity of the relu'd score function.**
> >
> > AR: The problem formalization provided in Section 3 is novel to this paper. Its key distinctions from prior work are a) the incorporation of an unobservable noise distribution into the final disease outcomes, and b) the construction of a set-based utility function which incentivizes diversity in the set of proposed candidate points, including the proposed score function in Eq. (4). Section 6 contrasts our problem formulation with that of prior work.
> > While the BAX framework is not new, our formulation of an algorithm which can leverage this framework to approximately optimize the objective from Section 3 is novel. In particular, by formulating a sub-modular score function, we are able to construct a computationally efficient greedy algorithm for which it is tractable to run a BAX execution procedure.
> >
> >
> > **C6: A minor (clarity) point, but the figures are poor quality. I cannot read the axis labels when printed out. Unfortunately this gives the impression of a rushed paper.**
> >
> > AR: Thank you for pointing this out. We have improved the axes on Figure 2 in the revised version. The other Figures in the main text and supplementary material are all reading properly when printing the paper on our end, but please let us know if there is any other table/figure with outstanding issues there.

---

### Official Review · Reviewer_3hGy · 2022-11-09

**Confidence:** 4
**Correctness:** 3
**Technical Novelty And Significance:** 2
**Empirical Novelty And Significance:** 2
**Recommendation:** 3

**Clarity, Quality, Novelty And Reproducibility:**

Quality:
- Overall the paper is well written; the problem motivation and validation part (esp. overfitting; see comments) could be strengthened. Figure numbering is broken.

Clarity:
- Text is easy to read and follow
- Some more intuitive descriptions of the algorithm could be added
- Figure numbering is broken.

Originality:
- The work provides a new solution to a previously established problem and claims performance gains
- The combination of problem and algorithmic solution seems to be new

Novelty And Reproducibility:
- I did not try to replicate the work but code is well organized and openly licensed, and seems robust. Some more guidance in the README landing page would be warranted.


**Strength And Weaknesses:**

Strengths:
+ probabilistic approach is neat for uncertainty characterization
+ practical combination of task, method derivation, and algorithm implementation
+ Openly licensed source code for the algorithms available

Weaknesses:
- Figure numberings are broken, text cites figure up to Fig.5, manuscript only has 3 figures, refs to Figs 3-4 are missing; not clear how to follow
- The problem motivation is hypothetical; it is not clear that "maximizing change" is what one would like to achieve in genomic drug discovery; on the other hand the algorithm is of interest regardless but perhaps best evaluated on its own right as a target optimization task.
- I am not sure if potential overfitting is sufficiently addressed in this work; explanation on this part could be strengthened

**Summary Of The Paper:**

This work presents a new probabilistic algorithm ("DiscoBAX") for subset selection that aims to approximately optimize phenotype movement in genomic intervention and can be useful in drug discovery tasks according to the authors. The method identifies a set of interventions whose elements will trigger maximum expected change in the disease outcome with respect to for some noise distribution. Performance is assessed based on synthetic data as well as public benchmarking data, and the algorithm is shown to outperform alternative approaches.



**Summary Of The Review:**

The work provides an interesting approach to an optimization problem that is initially motivated by certain genomic drug discovery tasks. Overall the reporting is clear and appears reproducible. The algorithmic innovation is in place but focuses on a relatively specific task definition where the applied motivation remains questionable, and it is not clear whether overfitting has been sufficiently controlled as the algorithm appears rather flexible. The paper has figure numbering problems.

---

> ### Author Response · Authors · 2022-11-19
> **Author Response to Official Review by Reviewer 3hGy (Part 1)**
>
> We thank the reviewer for their time reviewing our submission. After reading the review, we would like to underscore three critical points about our work that appear to have been misunderstood:
> 1. We do not introduce a “probabilistic algorithm for subset selection that aims to approximately optimize phenotype movement in genomic intervention” but rather devise a novel *acquisition function* in a Bayesian Algorithm Execution (BAX) setting to select the most relevant interventions (more on that below). This has important implications as it pertains to some of the points raised by the reviewer (see responses to C1 and C2 in particular);
> 2. This acquisition function promotes the selection of batches that *both* have high target values and are diverse. The notion of “diversity” is fully absent from the paper review, yet it is the most critical aspect of our work in terms of objectives and contributions;
> 3. Unlike what is stated by the reviewer, we do not seek to be "maximizing change" in the experimental phenotype but rather aim to select the optimal intervention set for the next stage of the experimental pipeline (ie., top hits with high phenotype response and high diversity) as is the case in the majority of works focusing on genomics for drug target identification (see details in C1).
>
> We provide detailed responses to all points raised by the reviewer below. Changes to the text (eg., clarifications, new experiments) are coloured red in the revised manuscript. In light of these responses and given the misunderstandings around important aspects of the work as listed above, we kindly ask the reviewer to reconsider their overall score. We are looking forward to engaging further during the discussion period to address any questions or concerns you may still have.
> (Cx: Reviewer’s Concern number x, AR: Author Response)
>
> C1: **The problem motivation is hypothetical; it is not clear that "maximizing change" is what one would like to achieve in genomic drug discovery; on the other hand the algorithm is of interest regardless but perhaps best evaluated on its own right as a target optimization task.**
>
> AR: We agree with the reviewer that “maximizing change” is not what one typically would like to achieve in genomic drug discovery and would like to reiterate that this is not what our approach intends to do either. Instead, we seek to identify the top interventions (ie., the top “hits”) which both have high phenotype change and diversity (see Figure 1). The confusion may stem from the fact that we used, in our mathematical formulation, a maximization over the expected measured phenotype (Eq. (4) in the revised manuscript) -- but this is rather a means to an end, and we do evaluate our methods relative to baselines (eg., in Table 1) in terms of their ability to select the most relevant interventions (ie., hits in terms of phenotype change and high diversity). We have also adjusted all sentences that could have been ambiguous about this point in the revised manuscript.
>
> C2: **I am not sure if potential overfitting is sufficiently addressed in this work; explanation on this part could be strengthened**
>
> AR: We kindly refer the reviewer to Section D.3 in the Appendix where steps towards overfitting mitigation with respect to the *acquisition function hyperparameters* are discussed. Furthermore, we note that in our setting where, at each cycle, a given acquisition function selects a batch of interventions based on a underlying model, any **model** overfitting would immediately impact the performance of all acquisition functions (ours and other baselines) and would be directly visible in the corresponding performance curves (eg., Figure 3). We prevented that from happening by closely following the experimental protocol in [1] and selected similar model architectures and hyperparameters. We added that last clarification in section Appendix D.2. of the revised manuscript.
>
> [1] Mehrjou, A., Soleymani, A., Jesson, A., Notin, P., Gal, Y., Bauer, S., & Schwab, P. (2022). GeneDisco: A Benchmark for Experimental Design in Drug Discovery. ICLR 2022.

---

> > ### Author Response · Authors · 2022-11-19
> > **Author Response to Official Review by Reviewer 3hGy (Part 2)**
> >
> > C3: **Figure numberings are broken, text cites figure up to Fig.5, manuscript only has 3 figures, refs to Figs 3-4 are missing; not clear how to follow**
> >
> > AR: Figures 4 and 5 (and 6 -- which was newly added during rebuttal) are all present in the Appendix. This is unfortunately a common bug with openreview submissions: the references broke when the paper was split into main text and Appendix (in supplementary) at submission time. We kindly ask the reviewer to disregard this as a particular issue of our paper as the problem will automatically be resolved when the two parts are combined again in the final version of the paper.
> >
> > C4: **Some more intuitive descriptions of the algorithm could be added**
> >
> > AR: As suggested by the reviewer, we added a new section in the Appendix of the revised manuscript (see section E & Figure 6) in which we include additional examples to further illustrate and provide intuition into the proposed algorithm (in particular Eq. (4) as per the revised manuscript).
> >
> > C5: **Some more guidance in the README landing page would be warranted.**
> >
> > AR: We added more thorough guidance on how to run experiments in the README page of the repository as suggested.

---

### Author Response · Authors · 2022-11-19
**Author Response to all Reviewers**

Dear Reviewers,

We would like to thank you for the time spent to engage with our paper and really appreciate the thoughtful comments. Based on your feedback, we have made several edits to the texts and conducted additional experiments as requested. We believe the submission is much stronger as a result. We summarize the key points of feedback and how we addressed them as follows:
1. **Clarifying technical novelty and contributions (reviewers JiqB and efcP)**: we re-arranged sections 2 (Background) and 4 (Method) to be clearer about what pertains to existing literature (eg., BAX) Vs what are new contributions of our work, and clarified technical contributions in responses below (see detailed responses to JiqB point C1 and efcP point C5)
2. **Comparison with additional Bayesian Optimization baselines (reviewer uzVy)**: we have added comparisons to the suggested baselines in the synthetic datasets and GeneDisco experiments
3. **Clarifying results in the experimental section (reviewers uzVy, JiqB and efcP)**: we improved the way results are presented (e.g., Table 1) and how conclusions connect to the initial objectives at the beginning of section 5
4. **Providing more intuition about the introduced algorithm (reviewers 3hGy and efcP)**: we added an illustrative example in Appendix E to provide the reader with more intuition into the inner workings of the proposed method
We are addressing all reviewer comments in detail below. We would be very happy to engage further during the rest of the discussion period to clear out any outstanding concerns or questions about the paper.

---

### Decision · Program_Chairs · 2023-01-20

**Decision:**

Reject

**Justification For Why Not Higher Score:**

While it is very commendable that the authors tried to revise and improve the paper during discussion, the overall concerns from the reviewers are not sufficiently addressed, but the paper has well-grounded motivation and interesting ideas that should be noted. It would be very good for the authors to further improve based on the reviewers' comments for future submission.

**Justification For Why Not Lower Score:**

NA

**Metareview: Summary, Strengths And Weaknesses:**

The authors in this work proposed a new method for defining the genetic interventions that could maximize the chance to direct a desirable phenotype in a complex biological system. The authors demonstrated their approach using synthetic as well as real world datasets to show good performance compared with benchmark. The overall concern is that the scope and the significance of the advancement of the work is limited.